# Feasibility of Shoulder Kinematics Assessment Using Magnetic Inertial Measurement Units in Hemiplegic Patients after Stroke: A Pilot Study

**Maria Longhi [1], Danilo Donati [1,2], Monica Mantovani [1], Silvia Casarotti [1], Lucia Calbucci [3], Giulia Puglisi [1], Daniela Platano [3,4,*] and Maria Grazia Benedetti [3,4]**

1   Rehabilitation Medicine, Department of Neuroscience, Azienda Ospedaliero-Universitaria di Modena, 41125 Modena, Italy; longhi.maria@aou.mo.it (M.L.); donati.danilo@aou.mo.it (D.D.)

2   Clinical and Experimental Medicine PhD Program, University of Modena and Reggio Emilia, 41121 Modena, Italy

3   Physical Medicine and Rehabilitation Unit, IRCCS Istituto Ortopedico Rizzoli, 40136 Bologna, Italy; mariagrazia.benedetti@ior.it (M.G.B.)

4   Department of Biomedical and Neuromotor Sciences, University of Bologna, 40126 Bologna, Italy

\*   Correspondence: daniela.platano@unibo.it; Tel.: +39-051-636-6355

**Abstract:** Scapulothoracic movements are altered after stroke, with resulting shoulder dysfunction. The scapulohumeral rhythm (SHR) is complex and poorly studied. Magnetic inertial measurement units (MIMUs) allow a rapid and accurate analysis of shoulder kinematics. MIMUs were used to assess the SHR during active shoulder flexion and abduction of over 60°. SHR values obtained from the hemiplegic shoulders of stroke patients (n = 7) were compared with those from healthy controls (n = 25) and correlated with clinical–functional measurements. The impairment of paretic arms was assessed using the Fugl-Meyer Assessment (FMA). We found that in paretic shoulders, the scapular tilt was significantly lower at maximal arm flexion and at 60° and 90° of arm abduction. On the paretic side, the SHR was also consistently lower for all measured arm movements. The FMA was correlated with the scapular anterior–posterior tilt at 60° and 90° of shoulder abduction (Rho = 0.847, $p = 0.016$, and Rho = 0.757, $p = 0.049$, respectively). This pilot study demonstrates the feasibility of MIMUs in assessing SHR in stroke patients and confirms previous findings on scapular dysfunction in stroke patients.

**Keywords:** shoulder; hemiplegia; kinematics; inertial sensor; scapula; stroke

## 1. Introduction

Shoulder impairment is a common complication after stroke, persisting in almost 60% of patients [1] at 12 weeks after stroke [2]. It is characterized by reduced glenohumeral motion, spasticity, subluxation, and somatosensory impairments in the paretic arm, contralateral to the brain lesion [3]. The non-paretic arm, ipsilateral to the cerebral lesion, has also been demonstrated to show strength and motor control impairments [4]. Abnormal active shoulder complex movement causes loss of function, limitation in activities of daily living [5], and a major issue for rehabilitation. Poor scapulothoracic positioning and altered scapulohumeral motion are considered risk factors in the development of shoulder dysfunction and pain after stroke [3].

Shoulder kinematics is the result of a complex motion that involves the movement of the glenohumeral joint and the movement of the scapula with respect to the thorax [3,5]. The coupling of these two movements during arm elevation has a precise ratio, described as the scapulohumeral rhythm (SHR).

Three-dimensional shoulder complex motion has been studied in patients with chronic stroke using different clinical and instrumental assessment measures. While clinical scores

are affected by subjectivity and inaccuracy in approaching diagnosis and are very time-consuming [6,7], instrumental measures provide objective measures. Specifically, wearable sensors, easy to use in a clinical setting, are gaining a certain popularity [6]. Other studies have addressed the instrumental assessment of shoulders using non-wearable devices such as optoelectronic systems, ultrasound-based systems, and electromagnetic tracking systems. These systems typically provide reliable measures, but they cannot be used for routine clinical assessment [3,6]. Most of the studies that have explored shoulder kinematics in post-stroke patients are based on instrumental 3D motion analysis [8–15].

The available studies based on the use of wearable devices in stroke patients [16] are mainly devoted to exercise monitoring (also remote) during rehabilitation [16–19] or the estimation of clinical scores for quality of movement (e.g., the Wolf Motor Function Test [17,20], Fugl-Meyer Assessment test [7,21,22], and Constant–Murley score [23]).

Several variables have also been explored, including shoulder motion coordination, smoothness, the presence of compensatory movements, speed, acceleration, and the amplitude of ROM [6,24]. However, many problems exist in these studies due to the variability in the instrument type, location of sensors, and motor task explored [16,24–26]. Only one study provides reference values for stroke shoulder flexion and abduction using a wearable device to monitor the effects of rehabilitative training [22]. Since effective rehabilitative treatment requires a good understanding of the mechanisms underlying shoulder impairment, a reliable quantitative assessment is relevant for clinical decision making and outcome measurement.

To evaluate the feasibility of measuring 3D shoulder complex kinematics (glenohumeral and scapulothoracic motion) in a clinical setting and during active flexion and abduction of the paretic arm in individuals with stroke, a validated inertial system was used. A post-stroke group of patients was compared with healthy controls for paretic and non-paretic upper limb motion. The kinematics obtained were then correlated with clinical–functional measurements.

## 2. Materials and Methods

### 2.1. Participants

A convenience sample of 16 consecutive individuals with sub-acute stroke (mean age 53.31 ± 13.25 years) undergoing rehabilitation on an outpatient basis at the rehabilitation medicine unit of the local hospital was consecutively recruited. The inclusion criteria for the stroke group were unilateral, ischemic, and/or hemorrhagic stroke; spasticity of shoulder muscles evaluated using the Modified Ashworth Scale (MAS) of less than 3; pain evaluated using the visual analog scale (VAS) of less than or equal to 7; and active shoulder flexion and abduction. The exclusion criteria were cognitive deficits in the mini–mental state examination (MMSE) of less than 4, age younger than 18, treatment with botulinum toxin within the last 4 months; bilateral, brainstem, or cerebellar stroke; shoulder pain before stroke; and orthopedic shoulder pathologies (complete rotator cuff tear, fractures, or previous shoulder surgery). De Baets et al. [8] stated that an active humerothoracic elevation of at least 60° is an absolute prerequisite to measure scapular behavior; therefore, only 7 patients with humeral flexion and abduction of greater than 60° were analyzed. Patients gave their consent to measure their shoulder motion with MIMUs, and they signed a written informed consent form. Experimental measurements and clinical–functional evaluations were performed at the rehabilitation outpatient gymnasium of Azienda Ospedaliero-Universitaria di Modena Hospital, Modena, Italy.

### 2.2. Clinical Evaluation of Function

Shoulder pain was assessed using the VAS, which rates pain from 0 to 10 (0 = no pain and 10 = the worst pain possible). Motor function of the paretic arm after stroke was evaluated using the FMA [27]. The total score for the shoulder and elbow subscale is 42, meaning normal use of the arm. In this study, we decided to use only the 5 items that evaluate the shoulder:

1. The hand from the contralateral knee to the ipsilateral ear;
2. Shoulder flexion at 180° (with the elbow at 0° and pronation–supination at 0°);
3. Shoulder abduction at 180° (elbow at 0° and forearm pronated);
4. The hand from the ipsilateral ear to the contralateral knee;
5. The hand to the lumbar spine from the hand on the lap.

The FMA items were scored 0–3 points. The maximum score was 15.

### 2.3. Scapular Kinematics

Wireless miniature inertial measurement units (MIMU, WISE, NCS Lab, Carpi, Italy) and the software SHoWlder (NCS Lab, Carpi, Italy) were used to evaluate 3D shoulder kinematics bilaterally. MIMU signals were recorded with a sample frequency of 60 Hz. Sensors were placed according to the ISEO-validated protocol [28]. Five MIMUs were fixed with skin tape: one to the center of the manubrium sternum, two on each suprascapular fossae at a medial distance between the medial portion of the acromion and the lateral part of the scapular spine, and two over the lateral aspect of both arms at a medial distance between the acromion process and the olecranum process, as shown in Figure 1. Patients stayed seated in a chair without supporting their backs, with the trunk upright, feet flat on the floor, humerus alongside the trunk, and elbows flexed at 90°. This position was maintained for a few seconds for calibration. Then, patients performed 5 repetitions of elevating both arms simultaneously to maximum and then lowering them to resting position in the sagittal plane (anterior flexion) and in the frontal plane (abduction), with the elbow fully extended and the thumb pointing up, and they were then recorded. The first and last repetitions were excluded, and only the mean of repetitions 2, 3, and 4 was considered.

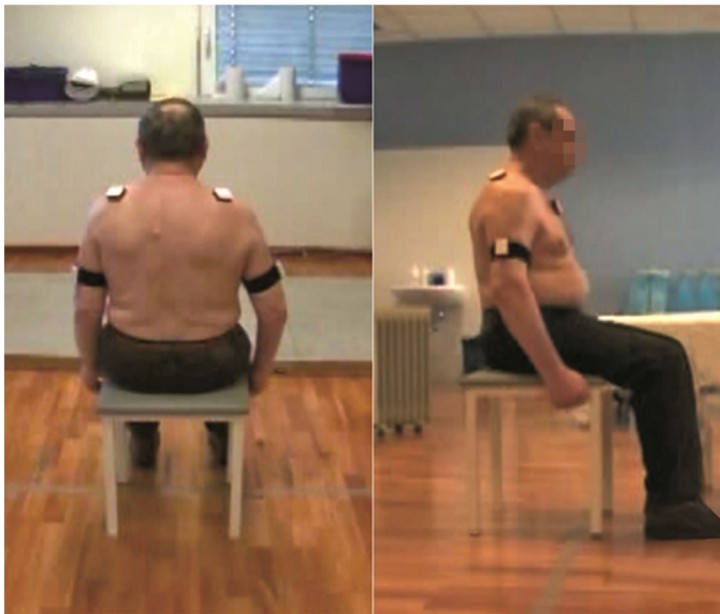

**Figure 1.** Five MIMUs were affixed: one to the center of the sternum, two between the medial portion of the acromion and the lateral part of the scapular spine, and two over the middle third of the arm.

Scapular movement on the thorax was recorded during humeral flexion and abduction with both arms: internal–external rotation, also referred to as protraction–retraction (PR), mediolateral upward/downward rotation (UD), and anterior–posterior tilt (TI). Protraction, upward rotation, and posterior tilt were positive values, while retraction, downward rotation, and anterior tilt were negative values. The SHR was measured for shoulder abduction and flexion.

*2.4. Data Analysis*

Data were expressed as means ± standard deviation (SD). The SHR was calculated in stroke patients on the paretic and the non-paretic sides. The Spearman coefficient was used to evaluate the correlation between the clinical FMA scores and instrumental data (maximum flexion and maximum abduction of the humerus; scapular PR, TI, UD in maximum flexion and maximum abduction of the humerus; and the SHR for PR, TI, UD at 30°, 60°, 90°, and 120° of flexion and abduction) for the 7 patients' paretic and non-paretic arms. The Wilcoxon non-parametric test, calculated using a Monte Carlo method for small samples, was used to evaluate the correlation between the paretic and non-paretic arm for the instrumental data. Data from 25 healthy controls, with a mean age of 37 ± 11 years [29], were reported as a reference for readers and compared with the stroke patients' data only qualitatively because it was not possible to proceed with statistical analysis. A statistical consultant from our institute performed a statistical analysis using the statistical package for social sciences (SPSS) software, version 15.0 (SPSS Inc., Chicago, IL, USA). A *p*-value of < 0.05 was considered significant for all analyses.

## 3. Results

*3.1. Demographic Characteristics*

Only one patient reported slight pain in their paretic shoulder (VAS = 3). FMA scores ranged from 8 to 15. Only one patient had spasticity in their shoulder muscles (grade 1) (Table 1).

**Table 1.** Patient clinical and general data.

| Pts | Age (Year) | Follow-Up (Months) | Paretic Side | FMA (0–15) | MAS | VAS |
|-----|-----------|--------------------|--------------|-----------|-----|-----|
| 1 | 47 | 1.5 | Left | 15 | 0 | 0 |
| 2 | 57 | 12.5 | Right | 12 | 0 | 0 |
| 3 | 58 | 36.4 | Right | 11 | 0 | 0 |
| 4 | 27 | 28.4 | Left | 10 | 1 | 0 |
| 5 | 74 | 61.8 | Right | 8 | 0 | 0 |
| 6 | 55 | 32.7 | Right | 10 | 0 | 3 |
| 7 | 58 | 13.8 | Left | 9 | 0 | 0 |

FMA = Fugl-Meyer Motor Assessment; MAS = Modified Ashworth Scale; VAS = visual analog scale.

*3.2. Glenohumeral Kinematics and Scapular Movements at Maximum Flexion and Extension*

The maximum shoulder RoM was lower for flexion on the paretic side (121.4° ± 18.7), with a significant difference (*p* = 0.032) compared with the non-paretic side (141.8° ± 17.9). Paretic shoulder abduction (139.7° ± 33.6) was also lower compared with the non-paretic side (171.3° ± 14.2), although not significantly, due to the large variance on the paretic side (*p* = 0.081) (Table 2). A significant difference was also found for scapular tilt during maximum shoulder flexion on the paretic side (17° ± 7.4), which was lower compared with the non-paretic side (25.6 ± 4.7) (*p* = 0.032). The non-paretic shoulders had a degree of maximum flexion and abduction close to that of the controls.

**Table 2.** Glenohumeral kinematics and scapular movements for paretic and non-paretic arms during maximum shoulder flexion and abduction. RoM values are reported as means (SD).

| | Paretic Side | Non-Paretic Side | Ruiz Ibán et al., 2020 [29] |
|---|--------------|------------------|------------------------------|
| RoM MaxFlex * | 121.4 (18.7) | 141.8 (17.9) | 137.6 (9.2) |
| RoM MaxAbd | 139.7 (33.6) | 171.3 (14.2) | 170.3 (13.4) |
| Scapular PR MaxFlex | 19.5 (7) | 17.8 (9.4) | |
| Scapular TI MaxFlex * | 17.0 (7.4) | 25.6 (4.7) | |
| Scapular UD MaxFlex | 33.3 (8.6) | 29.4 (8.7) | |
| Scapular PR MaxAbd | 15.5 (8.1) | 16.5 (6.8) | |
| Scapular TI MaxAbd | 17.1 (9.2) | 19.1 (8.9) | |
| Scapular UD MaxAbd | 29.4 (11.5) | 31.4 (9.7) | |

RoM = range of movement; Flex = flexion; Abd = abduction; PR = protraction–retraction; TI = anterior–posterior tilt; UD = upward/downward rotation. * *p* = 0.032 (paretic side vs. non-paretic side).

### 3.3. Scapular Angles during Different Angles of Shoulder Flexion and Abduction

The data regarding scapular angles during arm flexion and abduction at 30°, 60°, 90°, and 120° for the paretic and non-paretic sides and compared with healthy controls [29] are presented in Table 3. Significant differences between paretic and non-paretic values were found for scapular tilts of 60° and 90° during arm abduction due to greater motion on the non-paretic side ($p = 0.015$ and $p = 0.0017$, respectively).

**Table 3.** Scapular angles during shoulder flexion and abduction.

|  | Paretic Side | Non-Paretic Side | Ruiz Ibán et al., 2020 [29] |
|---|---|---|---|
| **30° flexion** | | | |
| PR | 2.3 (2.6) | 0.5 (1.6) | 1.4 (1.7) |
| UD | 5.0 (2) | 4.2 (2.2) | 2.6 (1.7) |
| TI | 2.4 (2.1) | 2.8 (3.2) | 2.5 (2.7) |
| **60° flexion** | | | |
| PR | 1.5 (5.3) | 0.9 (3.7) | 2.3 (2.8) |
| UD | 14.8 (4.8) | 12.8 (4.8) | 9.6 (3.1) |
| TI | 5.3 (6.2) | 6.1 (3.4) | 4.5 (4.2) |
| **90° flexion** | | | |
| PR | −3.3 (11.1) | −1.9 (5.3) | 1 (3.9) |
| UD | 23.7 (6.5) | 21.6 (6.5) | 18.6 (4.3) |
| TI | 10.0 (10.1) | 11.6 (3.5) | 8 (5.7) |
| **120° flexion** | | | |
| PR | −8.5 (8.5) | −7.3 (8.4) | −4.6 (5.2) |
| UD | 32.1 (3.1) | 25.8 (7.6) | 25.2 (5.4) |
| TI | 11.1 (13.7) | 18.9 (3.4) | 12.9 (6.6) |
| **30° abduction** | | | |
| PR | 1.7 (3.9) | −2.0 (5.1) | −1.7 (2.7) |
| UD | 9.1 (8.7) | 6.1 (5.8) | 3.7 (2.3) |
| TI | 2.3 (5.5) | 6.7 (3.6) | 1.9 (2.3) |
| **60° abduction** | | | |
| PR | 2.7 (6.8) | −2.9 (8.2) | −3.1 (3.9) |
| UD | 16.8 (10.4) | 12.9 (8.6) | 10.7 (3.4) |
| TI * | 3.5 (7.4) | 10.7 (4.9) | 4.1 (4) |
| **90° abduction** | | | |
| PR | 1.8 (10.4) | −3.3 (10.7) | −4 (4.8) |
| UD | 23.7 (8.8) | 17.1 (9.8) | 18.1 (4.1) |
| TI ** | 5.5 (10.5) | 14.0 (5.8) | 7.3 (5.3) |
| **120° abduction** | | | |
| PR | −6.3 (12.2) | −3.5 (11.5) | −3.5 (5.7) |
| UD | 32.5 (6.5) | 20.7 (10.8) | 24 (5.3) |
| TI | 7.6 (12.2) | 16.4 (6.4) | 10.3 (6.0) |

PR = protraction–retraction; UD = upward/downward rotation; TI = anterior–posterior tilt. * $p = 0.015$ and ** $p = 0.0017$ (paretic side vs. non-paretic side).

Figure 2 shows the scapulothoracic UD changes corresponding to arm movements. Major differences were found between the paretic and non-paretic sides during arm abduction, while during flexion, the scapulothoracic rotation was not particularly affected on the paretic side. The SHR was calculated as the ratio between the angle of shoulder abduction or flexion and the respective mean scapular upward/downward rotation angle for the paretic and non-paretic sides (Table 4). For shoulder abduction on the paretic side, the SHR ranged between 3.3 and 4.7, while it was slightly greater on the non-paretic side, ranging from 4.9 to 5.4. For shoulder flexion, there was an inverted trend on the paretic side, in which the ratio decreased from 6 to 3.6, and on the non-paretic side, from 7.2 to 4.8.

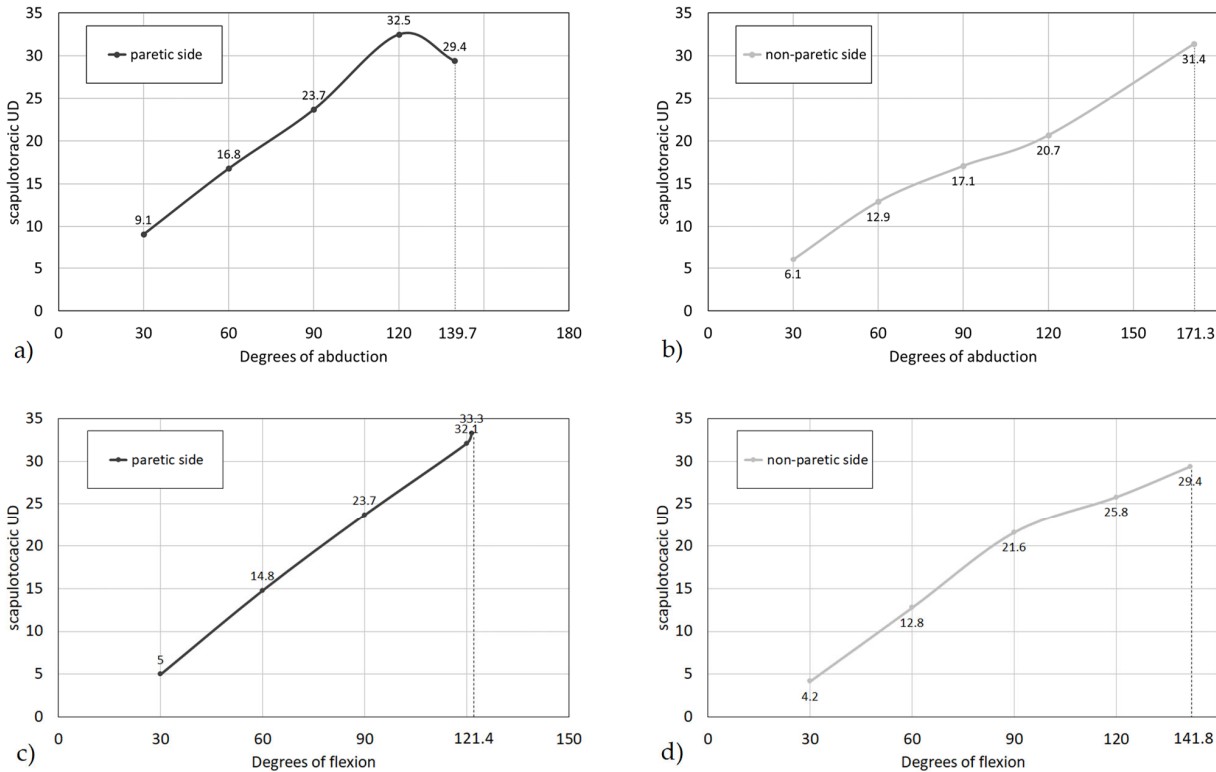

**Figure 2.** Top panels: scapulothoracic UD on the paretic side (**a**) vs. the non-paretic side (**b**) for different degrees of humeral abduction. Bottom panels: scapulothoracic UD on the paretic side (**c**) vs. the non-paretic side (**d**) for different degrees of humeral flexion.

**Table 4.** SHR during arm abduction and flexion.

|  | SHR Paretic Side | SHR Non-Paretic Side |
| --- | --- | --- |
| 30° abduction | 3.3 | 4.9 |
| 60° abduction | 3.6 | 4.7 |
| 90° abduction | 3.8 | 5.3 |
| 120° abduction | 3.7 | 5.8 |
| Max. abduction [#] | 4.7 | 5.4 |
| 30° flexion | 6.0 | 7.2 |
| 60° flexion | 4.1 | 4.7 |
| 90° flexion | 3.8 | 4.2 |
| 120° flexion | 3.7 | 4.6 |
| Max. flexion [§] | 3.6 | 4.8 |

[#] 139.7° for the paretic side and 171.3° for the non-paretic side; [§] 121.4° for the paretic side and 141.8° for the non-paretic side.

### 3.4. Correlation between Glenohumeral and Scapular Kinematics and FMA

A correlation was found between the FMA and scapular anterior–posterior tilt (TI) at maximum flexion of the humerus for the paretic side (Rho = 0.847; $p$ = 0.016). The FMA was also significantly correlated with scapular TI at 60° and 90° of shoulder abduction (Rho = 0.847, $p$ = 0.016, and Rho = 0.757, $p$ = 0.049, respectively).

### 4. Discussion

In this pilot study, we measured the glenohumeral motion and scapulothoracic angles during the shoulder flexion and abduction of paretic and non-paretic arms in a sample of post-stroke patients. The scapulohumeral rhythm was also calculated to shed light on the nature of shoulder dysfunction. Preliminary data were gathered from seven post-stroke patients to assess the feasibility of the wearable approach proposed by Cutti et al. [28].

This approach is very clinically oriented since it is valid and reliable when compared with a gold-standard optoelectronic system, and it is user-friendly and not time-consuming. Reference data exist [29], as well as other clinical applications in patients with rotator cuff tears [30].

The results show that patients with stroke presented a reduced scapular posterior tilt of the paretic shoulder during maximum flexion and during abduction at 60° and 90° compared with the non-paretic side. An abnormal scapular tilt was also correlated with the Fugl-Meyer score. Unfortunately, the lack of reference data on the kinematics of the shoulder using wearable devices made it impossible to compare the findings of the present pilot study. The only data available are from the study by Lin et al. [22] for the active range of motion of the shoulder in a group of 20 stroke patients before rehabilitation, measured using IMUs, but the details of the IMUs were not included in the study. The measured flexion of the paretic shoulder was 85–81° in the study and control group (with a large standard deviation of 42.6–36.9), respectively, and it was, on average, less than the humeral RoM we measured in our study.

However, comparisons can be made with studies carried out using non-wearable devices for 3D instrumental shoulder kinematics assessment, which is considered the gold standard [8,31]. Unfortunately, one problem encountered is that most of these studies differ in the explored shoulder motion (i.e., elevation in the sagittal plane, the frontal plane, the scapular plane, or a self-selected plane, or motion during activities of daily life) and in the examined arm compared with its control (i.e., paretic vs. non-paretic upper limb or dominant vs. non-dominant limb). Furthermore, some authors only published graphs and no relative data [11,13,15], therefore making it difficult to compare the studies [5].

The maximum flexion and abduction findings in the current study (121.4° and 139.7° for peak flexion and peak abduction, respectively) are not comparable to other studies where, usually, scapular motion was measured at fixed humerothoracic angles (30°, 45°, 60°, and 120°) and a maximum range of motion was not reported. Only Rundquist et al. [5] reported 130.5° of maximum humeral abduction in 16 post-stroke patients, while De Baets et al. [9] reported no significant differences in scapulothoracic joint RoM between individuals with stroke and controls. The range of motion of the non-paretic limbs was similar to that of controls, which is different from Meskers et al. [11], who found a lower value of maximum thoracohumeral elevation (125.7°). Furthermore, no specific abnormalities were found in scapulothoracic motion and SHR, which, in flexion and abduction, showed a linear ratio similar to those reported in the literature [23].

Causes of abnormal non-paretic upper limb kinematics have been previously discussed [11] in terms of neurological impairment due to stroke consequences on neural pathways in the contralateral hemisphere and biomechanical improper function, so the non-paretic upper limb should not be considered for comparison with the affected limb. Normally, in healthy subjects, as measured via 3D kinematic evaluation [10,12,32], during shoulder flexion, the scapula shows upward rotation and external rotation followed by internal rotation and posterior tilt, and upward, external rotation, and posterior tilt during abduction. The findings in the present study confirm the reduced posterior tilt observed by other authors [8,12,33] during arm elevation. However, they differ from the literature in which a diminished scapular protraction during elevation in the sagittal plane [11], more scapular lateral rotation during active abduction and flexion [13], and increased scapular upward rotation were found. Furthermore, increased scapular internal rotation during arm elevation in a self-selected plane was found by Lixandrao et al. [12].

Our data, even from a small cohort of stroke patients, show that there is reduced scapular tilt during flexion and abduction. Furthermore, if we consider the graphs showing the scapulothoracic UD changes, we see that at the highest degrees of abduction, there is inflexion of the lateral movement of the scapula on the thorax, which could contribute to the reduced range of motion of the shoulder.

Regarding the scapulohumeral rhythm, only a few authors calculated its value during wide shoulder elevation, concluding that, in some stroke patients, the SHR can be

the same in the non-paretic upper limb, while in others, there can be reduced scapular upward rotation, and in others, increased upward rotation [34]. Rundquist et al. [10] confirmed a lower non-linear SHR in the paretic upper limb. Additionally, McQuede and Smidt [35] demonstrated that, during dynamic humeral elevation, the scapulohumeral rhythm changes depend on the phase of elevation and do not have a fixed 2:1 ratio, depending on the type of arm elevation. Unfortunately, only the plot for SHR is usually shown in the literature, without providing data [5,23]. Only one study provides SHR values in stroke patients with an arm elevation of over 80° [35]. The ratio was reported to be approximately 7.9:1 for the first 26 degrees and increased to 3:1 between 104 and 130 degrees of elevation. The SHR values in the present study are in overall agreement with these findings, and at a higher RoM, the scapular contribution decreases. We calculated the SHR as a pure ratio between humeral motion and scapulothoracic rotation at discrete angles of elevation and abduction, while McQuade and Smidt [35] derived it as the slope of fitted continuous values of scapular UD rotation for different degrees of humeral elevation.

According to the literature, abnormalities in shoulder complex motion are correlated with the Fugl-Myers score [7,21]. This confirms the possible role of wearable devices in monitoring patients in a home setting, although specific studies should be planned to explore the relationship between wearable motion measures and shoulder section or even overall upper limb joint FMA scores.

The assessment of the individual components of the shoulder complex motion pattern in stroke patients is relevant for assessing the biomechanical impairment that underpins the abnormal kinematics for rehabilitation purposes. Intervention in the humerothoracic joint should be carried out and particular attention should be paid to the scapulothoracic motion. MIMUs seem able to measure the scapular angles and the scapulohumeral rhythm quickly and reliably, contributing to the study of the kinematics of the shoulder complex quickly, reliably, and objectively and revealing possible changes based on the therapeutic intervention.

Of course, an EMG of the muscles responsible for scapular dyskinesia could be of interest. Different findings have been reported on this issue, but there are still conflicting conclusions due to heterogenous muscle impairment in stroke patients (spasticity/flaccidity, impingement, etc.) [3]. The electromyographic evaluation of the scapular muscles should be performed in association with a kinematics assessment to investigate scapula dyskinesia to identify spastic muscles [36]. The significant reduction in the anterior–posterior tilt at a 60° and 90° arm abduction and at maximum arm flexion (versus the non-paretic and control from the literature [29]) that we report likely reflects an increase in the scapular anterior tilt. It is possible that increased scapular anterior tilt during abduction and flexion could be a consequence of increased muscular recruitment of the trapezius [37]. DeBates et al. found that during maximum flexion, there is an altered activation of the lower trapezius and infraspinatus to compensate for the altered activation of the serratus anterior and to correct the scapulothoracic movement [9]. During abduction, the middle deltoid and upper trapezius of stroke patients also have an altered pattern [4]. This has been interpreted as compensation for reduced deltoid force [38] during active glenohumeral abduction. In the literature, it was reported that impaired muscle recruitment of the trapezius, serratus anterior, and rhomboids in stroke patients [3,39] could lead to greater scapular internal rotation (in particular, in the anterior serratus and the middle and lower trapezius) and greater scapular anterior tilt (in particular, in the upper trapezius) [8,37]. As a consequence, scapular malalignment is responsible for alterations in biomechanics with reduced active and passive glenohumeral RoM [40]. Scapular movements in other directions did not seem to differ between the paretic and non-paretic sides, even if the small number of subjects in this study must be taken into account.

Additionally, pain can explain possible scapular dyskinesia and abnormal shoulder motion, as previously demonstrated [12,13]. In the current study, only one patient reported shoulder pain, although without particular restriction of movement. Therefore, it was not possible to form a conclusion.

Our results might have clinical relevance since, while the FMA gives a general overview of upper limb mobility, the instrumental assessment with the protocol used can provide specific information on scapular movement, which has the greatest impact on the altered humeral rhythm so as to provide indications for specific rehabilitation. In stroke patients, alteration of recruitment and control of the periscapular muscles (rhomboid muscles, serratus anterior, trapezius, pectoralis muscles, levator scapulae, and latissimus dorsi) lead to changes in the position and movement of the scapula, which are defined as dyskinesia [41]. It is involved in the onset of instability and pain of the hemiplegic shoulder (incidence 16–84%) [42]. Preventing or reducing shoulder dysfunction is very important for rehabilitation. This can be attained via an instrumental assessment.

The value of the present pilot study in stroke patients is in the feasibility of the use of a validated and reliable wearable system and software to test shoulder scapulothoracic and glenohumeral 3D kinematics, including SHR assessment. The small sample of patients, the lack of a matched control group, and the missing data on scapulothoracic and scapulohumeral value at rest are limitations of this study, making these findings non-generalizable. Furthermore, research with a larger number of hemiplegic patients, an adequate control group, and controlled clinical variables, like pain, spasticity, side, and follow-up, will be the subject of future studies.

## 5. Conclusions

The results obtained in the present study show that the paretic shoulders of post-stroke patients displayed (i) a decreased scapular posterior tilt during flexion and abduction of the paretic shoulder and (ii) a lower SHR during all measured arm movements. Moreover, we found (iii) that the FMA was correlated with the scapular anterior–posterior tilt at 60° and 90° of shoulder abduction. These findings are relevant from a rehabilitative point of view since they make it possible to understand the role of scapulothoracic motion in shoulder kinematics and, possibly, to intervene with treatments for restoring muscle tone around the scapula.

Clinical scores normally used to evaluate function give information correlated with kinematics but are time-consuming and not sensitive to scapulothoracic motion. MIMUs allow for rapid, accurate, and ecological analysis of shoulder movement, succeeding in highlighting scapular rotation in relation to the flexion and abduction of the shoulder. In the literature, wearable devices have mainly been used to monitor the effects of rehabilitation on hemiplegic shoulders, while clinical decision making requires an objective assessment of shoulder impairment. The present pilot study described a validated and reliable wearable system to test shoulder scapulothoracic and glenohumeral 3D kinematics, including SHR assessment in stroke patients.

The scope of this study was to evaluate the feasibility of measuring 3D shoulder complex kinematics (glenohumeral and scapulothoracic motion) during the active flexion and abduction of the paretic arm in individuals with stroke using MIMUs, compare it with healthy patients, and correlate instrumental measurements with clinical–functional measurements.

Our study has some limitations: a control group without matching age to the post-stroke group; the inclusion of patients with different arm function; the small sample and the heterogeneous onset of stroke (subacute and chronic); and the lack of an electromyographic evaluation performed in association with the kinematics assessment.

A future study of paretic arm shoulder function in stroke should be carried out with a larger number of hemiplegic patients, an adequate control group, and controlled clinical variables in order to permit a stronger statistical analysis. Moreover, the combination of an electromyographic evaluation to identify specific muscle dyskinesia will help in the study and treatment of hemiplegic shoulders.

**Author Contributions:** Conceptualization: M.L. and M.G.B.; methodology: M.L. and M.G.B.; formal analysis: M.G.B.; investigation: M.L.; data Collection: M.M., S.C. and G.P.; writing—original draft

preparation: M.L. and D.D.; writing—review and editing: M.L., L.C., M.G.B. and D.P.; visualization. D.P.; supervision: M.G.B. All authors have read and agreed to the published version of the manuscript.

**Funding:** This research received no external funding.

**Institutional Review Board Statement:** This study was conducted according to the guidelines of the Declaration of Helsinki. As a pilot study for feasibility, no approval by the local Ethics Committee was requested. The data obtained were used to write a research protocol to be submitted to the Ethics Committee.

**Informed Consent Statement:** Informed consent was obtained from all subjects involved in this study.

**Data Availability Statement:** The data supporting the reported results can be obtained from the corresponding author upon reasonable request.

**Acknowledgments:** We would like to acknowledge and thank Marco Muraccini, Antonella Bearardi, Francesco Menon, and Matteo Mantovani of the NCS Lab., Carpi (MO), Italy, for their technical support in collecting data, Elettra Pignotti Stat. Eng. for the statistical analyses, and Andrea Cutti Eng. for their useful comments.

**Conflicts of Interest:** The authors declare no conflict of interest.

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
