# Peer review of "Feasibility of Shoulder Kinematics Assessment Using Magnetic Inertial Measurement Units in Hemiplegic Patients after Stroke: A Pilot Study"

_applsci, doi:10.3390/app132111900_

Round 1
Reviewer 1 Report
Comments and Suggestions for Authors
The aim of the study was to evaluate SHR in stroke patients using MIMUs in comparison to healthy patients 16 and correlate instrumental measurements to clinical-functional measurements.The research work is of great significance. But there are still some problems in the article that need to be revised.
1.The abstract of the article needs to be condensed.
2.1. A deeper bibliographic analysis is required in order to give a complete understanding of the state of the art. It is suggested to add these papers (and more):
Cai Z, Xia Y, Huang X. Analyses of pedestrian’s head-to-windshield impact biomechanical responses and head injuries using a head finite element model[J]. Journal of Mechanics in Medicine and Biology, 2020, 20(01): 1950063.
3.The patient's experimental location should be agreed by the Ethics Committee and should be stated in the text.
4.The discussion section should be further deepened
5.The conclusion should be described point by point
Comments on the Quality of English LanguageMinor editing of English language required.
Reviewer 2 Report
Comments and Suggestions for Authors
The topic is promising and the results are congruent. However, it is suggested the following:
1. The abstract section mentions the problem background and proposal solution, methodology and conclusions, but it needs to write, after methodology, the results with metrics (correlations).
2. To aggregate the equations and describe the calculus to get the correlations.
3. The Conclusion section must show, clearly the contribution of the article in comparison with the state of the art, scope, limits according to conditions of tests, applications of this solution (Feasibility of shoulder kinematics assessment using MIMUs...), and future works.
Comments on the Quality of English LanguageMinor editing of English language required
Reviewer 3 Report
Comments and Suggestions for Authors
I would modifie line 108-109 to be more exact on the placement of the MIMU concerning the 2 between medial port of acromion and lat part ( write f.ex. medial distance between the two points ...)
